# Salicylic Acid and Risk of Colorectal Cancer: A Two-Sample Mendelian Randomization Study

**DOI:** 10.3390/nu13114164

**Published:** 2021-11-21

**Authors:** Aayah Nounu, Rebecca C. Richmond, Isobel D. Stewart, Praveen Surendran, Nicholas J. Wareham, Adam Butterworth, Stephanie J. Weinstein, Demetrius Albanes, John A. Baron, John L. Hopper, Jane C. Figueiredo, Polly A. Newcomb, Noralane M. Lindor, Graham Casey, Elizabeth A. Platz, Loïc Le Marchand, Cornelia M. Ulrich, Christopher I. Li, Fränzel J. B. van Dujinhoven, Andrea Gsur, Peter T. Campbell, Víctor Moreno, Pavel Vodicka, Ludmila Vodickova, Efrat Amitay, Elizabeth Alwers, Jenny Chang-Claude, Lori C. Sakoda, Martha L. Slattery, Robert E. Schoen, Marc J. Gunter, Sergi Castellví-Bel, Hyeong-Rok Kim, Sun-Seog Kweon, Andrew T. Chan, Li Li, Wei Zheng, D. Timothy Bishop, Daniel D. Buchanan, Graham G. Giles, Stephen B. Gruber, Gad Rennert, Zsofia K. Stadler, Tabitha A. Harrison, Yi Lin, Temitope O. Keku, Michael O. Woods, Clemens Schafmayer, Bethany Van Guelpen, Steven Gallinger, Heather Hampel, Sonja I. Berndt, Paul D. P. Pharoah, Annika Lindblom, Alicja Wolk, Anna H. Wu, Emily White, Ulrike Peters, David A. Drew, Dominique Scherer, Justo Lorenzo Bermejo, Hermann Brenner, Michael Hoffmeister, Ann C. Williams, Caroline L. Relton

**Affiliations:** 1Integrative Cancer Epidemiology Programme (ICEP), Medical Research Council (MRC) Integrative Epidemiology Unit, Bristol Medical School, University of Bristol, Bristol BS8 2BN, UK; rebecca.richmond@bristol.ac.uk (R.C.R.); caroline.relton@bristol.ac.uk (C.L.R.); 2School of Cellular and Molecular Medicine, University of Bristol, Bristol BS8 1TD, UK; ann.c.williams@bristol.ac.uk; 3MRC Epidemiology Unit, School of Clinical Medicine, University of Cambridge, Cambridge CB2 0SL, UK; Isobel.Stewart@mrc-epid.cam.ac.uk (I.D.S.); nick.wareham@mrc-epid.cam.ac.uk (N.J.W.); 4British Heart Foundation Cardiovascular Epidemiology Unit, Department of Public Health and Primary Care, University of Cambridge, Cambridge CB1 8RN, UK; praveen.x.surendran@gsk.com (P.S.); asb38@medschl.cam.ac.uk (A.B.); 5British Heart Foundation Centre of Research Excellence, Division of Cardiovascular Medicine, University of Cambridge, Cambridge CB2 0QQ, UK; 6Health Data Research UK Cambridge, Wellcome Genome Campus and University of Cambridge, Cambridge CB10 1SA, UK; 7Department of Public Health and Primary Care, University of Cambridge, Cambridge CB1 8RN, UK; pp10001@medschl.cam.ac.uk; 8National Institute for Health Research Blood and Transplant Research Unit in Donor Health and Genomics, University of Cambridge, Cambridge CB2 1TN, UK; 9National Institute for Health Research Cambridge Biomedical Research Centre, Cambridge Biomedical Campus, University of Cambridge, Cambridge University Hospitals, Cambridge CB2 0QQ, UK; 10Division of Cancer Epidemiology and Genetics, National Cancer Institute, National Institutes of Health, Bethesda, MD 20814, USA; weinstes@mail.nih.gov (S.J.W.); albanesd@mail.nih.gov (D.A.); berndts@mail.nih.gov (S.I.B.); 11Department of Medicine, School of Medicine, University of North Carolina, Chapel Hill, NC 27516, USA; john_baron@med.unc.edu; 12Centre for Epidemiology and Biostatistics, Melbourne School of Population and Global Health, The University of Melbourne, Melbourne, VIC 3053, Australia; johnlh@unimelb.edu.au (J.L.H.); graham.giles@cancervic.org.au (G.G.G.); 13Department of Epidemiology, Institute of Health and Environment, School of Public Health, Seoul National University, Seoul 08826, Korea; 14Department of Medicine, Samuel Oschin Comprehensive Cancer Institute, Cedars-Sinai Medical Center, Los Angeles, CA 90048, USA; jane.figueiredo@cshs.org; 15Department of Preventive Medicine, Keck School of Medicine, University of Southern California, Los Angeles, CA 90032, USA; 16Public Health Sciences Division, Fred Hutchinson Cancer Research Center, Seattle, WA 98109-1024, USA; pnewcomb@fredhutch.org (P.A.N.); cili@fhcrc.org (C.I.L.); lori.sakoda@kp.org (L.C.S.); 17School of Public Health, University of Washington, Seattle, WA 98195, USA; 18Department of Health Science Research, Mayo Clinic, Scottsdale, AZ 85259, USA; nlindor@mayo.edu; 19Center for Public Health Genomics, Department of Public Health Sciences, University of Virginia, Charlottesville, VA 22908, USA; gc8r@virginia.edu; 20Department of Epidemiology, Johns Hopkins Bloomberg School of Public Health, Baltimore, MD 21205, USA; eplatz@jhsph.edu; 21Cancer Epidemiology Program, University of Hawaii Cancer Center, Honolulu, HI 96813, USA; loic@cc.hawaii.edu; 22Huntsman Cancer Institute, Department of Population Health Sciences, University of Utah, Salt Lake City, UT 84112, USA; neli@hci.utah.edu; 23Division of Human Nutrition and Health, Department of Agrotechnology and Food Sciences, Wageningen University & Research, 6700 HB Wageningen, The Netherlands; franzel.vanduijnhoven@wur.nl (F.J.B.v.D.); tharriso@fredhutch.org (T.A.H.); ylin2@fredhutch.org (Y.L.); ewhite@fredhutch.org (E.W.); upeters@fredhutch.org (U.P.); 24Institute of Cancer Research, Department of Medicine I, Medical University Vienna, 1090 Vienna, Austria; andrea.gsur@meduniwien.ac.at; 25Department of Population Science, American Cancer Society, Atlanta, GA 30303, USA; Peter.Campbell@cancer.org; 26Oncology Data Analytics Program, Catalan Institute of Oncology-IDIBELL, 08908 Barcelona, Spain; v.moreno@iconcologia.net; 27CIBER Epidemiología y Salud Pública (CIBERESP), 28029 Madrid, Spain; 28Department of Clinical Sciences, Faculty of Medicine, University of Barcelona, 08007 Barcelona, Spain; 29ONCOBEL Program, Bellvitge Biomedical Research Institute (IDIBELL), 08908 Barcelona, Spain; 30Department of Molecular Biology of Cancer, Institute of Experimental Medicine of the Czech Academy of Sciences, 142 20 Prague, Czech Republic; pvodicka@biomed.cas.cz (P.V.); ludovod@seznam.cz (L.V.); 31Institute of Biology and Medical Genetics, First Faculty of Medicine, Charles University, Nové Město, 121 08 Prague, Czech Republic; 32Faculty of Medicine and Biomedical Center in Pilsen, Charles University, 323 00 Pilsen, Czech Republic; 33Division of Clinical Epidemiology and Aging Research, German Cancer Research Center (DKFZ), 69120 Heidelberg, Germany; e.amitay@dkfz-heidelberg.de (E.A.); elizabeth.alwers@dkfz-heidelberg.de (E.A.); 34Division of Cancer Epidemiology, German Cancer Research Center (DKFZ), 69120 Heidelberg, Germany; j.chang-claude@dkfz.de (J.C.-C.); h.brenner@dkfz.de (H.B.); m.hoffmeister@dkfz.de (M.H.); 35Department of Oncology, Haematology and BMT, University Medical Centre Hamburg-Eppendorf, University Cancer Centre Hamburg (UCCH), 20251 Hamburg, Germany; 36Division of Research, Kaiser Permanente Northern California, Oakland, CA 94612, USA; 37Department of Internal Medicine, University of Utah, Salt Lake City, UT 84112, USA; marty.slattery@hsc.utah.edu; 38Department of Medicine and Epidemiology, University of Pittsburgh Medical Center, Pittsburgh, PA 15261, USA; rschoen@pitt.edu; 39Nutrition and Metabolism Section, International Agency for Research on Cancer, World Health Organization, 69372 Lyon, France; GunterM@iarc.fr; 40Gastroenterology Department, Hospital Clínic, Institut d’Investigacions Biomèdiques August Pi i Sunyer (IDIBAPS), Centro de Investigación Biomédica en Red de Enfermedades Hepáticas y Digestivas (CIBEREHD), University of Barcelona, 08036 Barcelona, Spain; sbel@clinic.cat; 41Department of Surgery, Chonnam National University Hwasun Hospital and Medical School, Hwasun 58128, Korea; drkhr@jnu.ac.kr; 42Department of Preventive Medicine, Chonnam National University Medical School, Gwangju 61186, Korea; ss.kweon2@gmail.com; 43Jeonnam Regional Cancer Center, Chonnam National University Hwasun Hospital, Hwasun 58128, Korea; 44Division of Gastroenterology, Massachusetts General Hospital and Harvard Medical School, Boston, MA 02114, USA; achan@partners.org; 45Channing Division of Network Medicine, Brigham and Women’s Hospital and Harvard Medical School, Boston, MA 02115, USA; DADREW@mgh.harvard.edu; 46Clinical and Translational Epidemiology Unit, Massachusetts General Hospital and Harvard Medical School, Boston, MA 02114, USA; 47Broad Institute of Harvard and MIT, Cambridge, MA 02142, USA; 48Department of Epidemiology, Harvard T.H. Chan School of Public Health, Harvard University, Boston, MA 02115, USA; 49Department of Immunology and Infectious Diseases, Harvard T.H. Chan School of Public Health, Harvard University, Boston, MA 02115, USA; 50Department of Family Medicine, University of Virginia, Charlottesville, VA 22903, USA; ll134q@rocketmail.com; 51Division of Epidemiology, Department of Medicine, Vanderbilt-Ingram Cancer Center, Vanderbilt Epidemiology Center, Vanderbilt University School of Medicine, Nashville, TN 37232, USA; wei.zheng@Vanderbilt.Edu; 52Leeds Institute of Cancer and Pathology, School of Medicine, University of Leeds, Leeds LS2 9JT, UK; d.t.bishop@leeds.ac.uk; 53Colorectal Oncogenomics Group, Department of Clinical Pathology, The University of Melbourne, Parkville, VIC 3010, Australia; daniel.buchanan@unimelb.edu.au; 54Melbourne Medical School, University of Melbourne Centre for Cancer Research, Victorian Comprehensive Cancer Centre, Parkville, VIC 3010, Australia; 55Genetic Medicine and Family Cancer Clinic, The Royal Melbourne Hospital, Parkville, VIC 3000, Australia; 56Cancer Epidemiology Division, Cancer Council Victoria, Melbourne, VIC 3004, Australia; 57Precision Medicine, School of Clinical Sciences at Monash Health, Monash University, Clayton, VIC 3168, Australia; 58Department of Preventive Medicine & USC Norris Comprehensive Cancer Center, Keck School of Medicine, University of Southern California, Los Angeles, CA 90033, USA; sgruber@coh.org; 59Department of Community Medicine and Epidemiology, Lady Davis Carmel Medical Center, Haifa 3448516, Israel; rennert@tx.technion.ac.il; 60Ruth and Bruce Rappaport Faculty of Medicine, Technion-Israel Institute of Technology, Haifa 3200003, Israel; 61Clalit National Cancer Control Center, Haifa 3436212, Israel; 62Memorial Sloan Kettering Cancer Center, Department of Medicine, New York, NY 10065, USA; stadlerz@mskcc.org; 63Center for Gastrointestinal Biology and Disease, School of Medicine, University of North Carolina, Chapel Hill, NC 27599-7555, USA; temitope_keku@med.unc.edu; 64Discipline of Genetics, Faculty of Medicine, Memorial University of Newfoundland, St. John’s, NL A1B 3V6, Canada; mwoods@mun.ca; 65Department of General Surgery, University Hospital Rostock, 18057 Rostock, Germany; Clemens.Schafmayer@uksh.de; 66Department of Radiation Sciences, Oncology Unit, Umeå University, 901 87 Umeå, Sweden; bethany.vanguelpen@umu.se; 67Wallenberg Centre for Molecular Medicine, Department of Biomedical and Clinical Sciences, Umeå University, 901 87 Umeå, Sweden; 68Lunenfeld Tanenbaum Research Institute, Mount Sinai Hospital, Faculty of Medicine, University of Toronto, Toronto, ON M5G 1X5, Canada; Steven.Gallinger@uhn.on.ca; 69Division of Human Genetics, Department of Internal Medicine, The Ohio State University Comprehensive Cancer Center, Columbus, OH 43210, USA; Heather.Hampel@osumc.edu; 70Department of Clinical Genetics, Karolinska University Hospital, 171 64 Solna, Sweden; Annika.Lindblom@ki.se; 71Department of Molecular Medicine and Surgery, Karolinska Institutet, 171 64 Solna, Sweden; 72Institute of Environmental Medicine, Karolinska Institutet, 171 64 Solna, Sweden; Alicja.Wolk@ki.se; 73Department of Surgical Sciences, Uppsala University, 751 85 Uppsala, Sweden; 74Department of Population and Public Health Sciences, Keck School of Medicine, University of Southern California, Los Angeles, CA 90032, USA; annawu@usc.edu; 75Department of Epidemiology, University of Washington School of Public Health, Seattle, WA 98195, USA; 76Institute of Medical Biometry and Informatics, Medical Faculty, University of Heidelberg, Im Neuenheimer Feld 130.3, 69120 Heidelberg, Germany; Dominique.Scherer@uni-heidelberg.de (D.S.); lorenzo@imbi.uni-heidelberg.de (J.L.B.); 77Division of Preventive Oncology, German Cancer Research Center (DKFZ), National Center for Tumor Diseases (NCT), 69120 Heidelberg, Germany; 78German Cancer Consortium (DKTK), German Cancer Research Center (DKFZ), 69120 Heidelberg, Germany

**Keywords:** salicylic acid, aspirin, colorectal cancer, Mendelian randomization

## Abstract

Salicylic acid (SA) has observationally been shown to decrease colorectal cancer (CRC) risk. Aspirin (acetylsalicylic acid, that rapidly deacetylates to SA) is an effective primary and secondary chemopreventive agent. Through a Mendelian randomization (MR) approach, we aimed to address whether levels of SA affected CRC risk, stratifying by aspirin use. A two-sample MR analysis was performed using GWAS summary statistics of SA (INTERVAL and EPIC-Norfolk, N = 14,149) and CRC (CCFR, CORECT, GECCO and UK Biobank, 55,168 cases and 65,160 controls). The DACHS study (4410 cases and 3441 controls) was used for replication and stratification of aspirin-use. SNPs proxying SA were selected via three methods: (1) functional SNPs that influence the activity of aspirin-metabolising enzymes; (2) pathway SNPs present in enzymes’ coding regions; and (3) genome-wide significant SNPs. We found no association between functional SNPs and SA levels. The pathway and genome-wide SNPs showed no association between SA and CRC risk (OR: 1.03, 95% CI: 0.84–1.27 and OR: 1.08, 95% CI: 0.86–1.34, respectively). Results remained unchanged upon aspirin use stratification. We found little evidence to suggest that an SD increase in genetically predicted SA protects against CRC risk in the general population and upon stratification by aspirin use.

## 1. Introduction

Colorectal cancer (CRC) is the fourth most common cancer in the UK and worldwide [1,2]. Although incidence rates among the over 50s have remained relatively stable, rates in younger age groups have increased in both the UK and US populations [3,4]. This highlights a need to find better and complementary prevention strategies to reduce risk of cancer. 

Salicylic acid (SA) is a dietary metabolite that can be found in various fruits, vegetables, herbs, and spices [5,6,7]. Results from a meta-analysis of 19 cohort studies found that combined intake of fruits and vegetables reduced the risk of colorectal cancer (summary relative risk (RR): 0.90, 95% CI: 0.83–0.98) [8]. Whilst dietary fibre obtained from fruits and vegetables indicates a possible mechanism for decreased risk [9], it has also been suggested that increased levels of SA obtained through their consumption may play a role [7]. In addition, salicylates can be obtained through pharmacological intervention in the form of aspirin (acetylsalicylic acid), a well-known analgesic used to treat fever, inflammation, and acute pain [10], which is rapidly deacetylated to form SA [11,12] (Figure 1), the active form of the aspirin metabolic pathway [13,14]. Whilst SA can be obtained from the diet, the concentrations achieved (male and female median intake from diet 4.4 mg/day and 3.2 mg/day, respectively [6]) are much lower than through aspirin ingestion (aspirin doses ranging between 75 mg to ≥325 mg given daily/alternate days) [15]. Therefore, it is unclear whether concentrations achieved from the diet alone are sufficient to protect against cancer or whether larger doses obtained through pharmacological intervention are required.

As of yet, no primary prevention trials have been carried out to assess the effect of SA intervention on CRC risk, but the evidence of aspirin as a chemopreventive agent is clear [17]. A long-term follow up of a randomised controlled trial (RCT) in the Women’s Health Study (WHS) showed that alternate day aspirin intake reduced the risk of CRC after a median of 17.5 years follow up (HR: 0.80, 95% confidence intervals (CI): 0.67–0.97) [18] and a meta-analysis of observational studies showed that aspirin is protective against CRC (relative risk (RR): 0.79, 95% CI: 0.74–0.85) [19]. Further evidence comes from RCTs for primary and secondary prevention of vascular events. These showed that aspirin reduces the risk of CRC incidence and mortality (HR: 0.76, 95% CI: 0.60–0.96 and odds ratio (OR): 0.79, 95% CI: 0.68–0.92, respectively) [20,21]. Considering aspirin is rapidly deacetylated to form SA in under 30 min [22], and that evidence in the form of in vivo and in vitro experiments have previously shown SA to be an antiproliferative and antitumour agent [23,24,25], it may be that metabolism of aspirin leading to increased circulating SA levels may partially explain aspirin’s chemopreventive mode of action.

Although many observational studies have shown an inverse association between aspirin use and CRC risk, few have directly assessed the association between SA itself and CRC. In order to identify the true effect of SA on CRC risk, conducting an RCT would be the ideal study design. However, RCTs for cancer primary prevention are lengthy and costly, therefore it would be helpful to test this association using statistical methods such as Mendelian Randomization (MR). MR uses genetic variants (mostly single nucleotide polymorphisms (SNPs)) related to modifiable factors (such as metabolite levels) to investigate the causal role of these factors on risk of disease [26,27,28]. Through this method, MR has been likened to RCTs in that genetic variants are randomly allocated at conception the same way that an intervention is randomly allocated at the start of a trial [29,30]. This lends many advantages such as overcoming the issues of confounding and reverse causation, which are commonly encountered in observational epidemiology [29]. MR has previously been useful in predicting trial outcomes such as the case of selenium and prostate cancer in The Selenium and Vitamin E Cancer Prevention Trial (SELECT) [31]. Results from an MR study mimicked the findings of this RCT and may have been useful to inform whether to conduct a trial that cost $114 million and that was weakly associated with increasing high-grade prostate cancer risk [32].

For this reason, we applied an MR approach using genetic “instruments” or proxies for SA to assess the causal effect of this metabolite on the risk of CRC. Since aspirin is rapidly deacetylated to SA [22], and therefore a plausible proxy of increased SA levels, we also stratified our analysis between aspirin users and non-users to test the hypothesis of whether diet-derived levels of SA alone would affect risk of CRC or whether higher concentrations achieved through pharmacological intervention in the form of aspirin was required to identify an effect. Based on previous observational evidence, we hypothesise that a genetically predicted increase in SA levels would reduce the risk of CRC, with a stronger effect observed in aspirin users.

## 2. Materials and Methods

### 2.1. Genetic Variants for Salicylic Acid

We applied a two-sample MR study design to test for the association of SA levels (sample 1) with risk of CRC (sample 2). GWAS and meta-analysis of salicylate levels were performed using 5841 participants from the EPIC-Norfolk study [33] and 8455 from the INTERVAL study [34]. The percentage of samples with missing salicylate measurements was low (0.43% and 1.44% in EPIC-Norfolk and INTERVAL respectively), providing a total sample size of 14,149. Salicylate was measured independently in each study as one of many metabolites measured using the Metabolon DiscoveryHD4^®^ platform (Metabolon, Inc., Durham, NC, USA), from non-fasted plasma samples (predominantly non-fasted samples in EPIC-Norfolk) collected at baseline. Measures that were median normalised for run day were natural log transformed, winsorised to 5 standard deviations from the mean, before being regressed against age, sex, and study-specific variables (measurement consignment in EPIC-Norfolk and measurement consignment, INTERVAL centre, plate number, appointment month, lag time between blood donation appointment and sample processing, and the first 5 ancestry principal components in INTERVAL) using linear regression. Residuals from this regression were standardised (mean 0, standard deviation 1) and used for further analysis. Genotyping was performed in both studies using the Affymetrix Axiom UK Biobank genotyping array. In INTERVAL, genotype imputation was performed using the combined UK10K+1000 Genomes Phase 3 reference panel. In EPIC-Norfolk, imputation was performed using the Haplotype Reference Consortium reference panel, with additional variants imputed using the UK10K+1000 Genomes Phase 3 reference panel. Genome-wide association analyses were performed using BOLT-LMM (version 2.2) [35], and variants with a MAF < 0.01% and INFO < 0.3 were excluded. Associations from the two studies were pooled using inverse variance weighted fixed effect meta-analysis implemented in METAL [36], applying a minor allele count threshold in each study of >10. 

The causal effect of SA on risk of CRC was assessed using 3 sets of genetic variants (SNPs) related to SA: (1) functional SNPs that influence aspirin and SA metabolising enzymes’ activity (derived from Figure 1)—termed “functional SNPs”; (2) pathway SNPs, those that are present in the coding regions of genes that are involved in aspirin and SA metabolism (based on the NCBI Build 37/UCSC hg19 from https://grch37.ensembl.org/index.html (accessed on 14 December 2016), Appendix A) termed “pathway SNPs”; (3) genome-wide significant SNPs associated with levels of circulating aspirin metabolites—termed “genome-wide SNPs”. Pathways SNPs were defined as having a Bonferroni threshold of association (*p* value 0.05/2701 = 1.85 × 10^−5^), a MAF ≥ 0.01%, as well as a consistent direction of effect in both EPIC-Norfolk and INTERVAL. Genome-wide signals were defined as having an association *p* value < 5 × 10^−8^ in the meta-analysis, a MAF ≥ 0.01%, consistent direction of effect across the two studies, and association *p* value < 0.01 in both studies

To account for genetic correlation, linkage disequilibrium (LD) clumping at an R^2^ < 0.001 and 10,000 kb window was performed to retain the SNP most strongly associated with the metabolite for downstream analysis. Since an R^2^ < 0.001 is considered highly stringent, we also used an R^2^ < 0.8 to incorporate more variants while accounting for residual correlation in the model (see Statistical Analysis). An F-statistic for each SNP–exposure association was calculated to reflect the strength of the genetic instrument and indicate any possibility of weak instrument bias, usually inferred when F < 10 [37]. Power calculations were conducted using the mRnd online calculator to identify the OR in both directions that could be detected with the sample size available [38].

### 2.2. Genetic Variants for CRC Incidence

SNP–outcome associations were obtained from the Colon Cancer Family Registry (CCFR), Colorectal Cancer Transdisciplinary Study (CORECT), Genetics and Epidemiology of Colorectal Cancer (GECCO) consortia, and the UK Biobank (55,168 cases and 65,160 controls), hereafter collectively termed as GECCO [39,40,41]. Genetic data from a population-based case-control study from southwestern Germany (Darmkrebs: Chancen der Verhütung durch Screening (DACHS)) was used to assess replication of the findings, and to run an MR analysis stratified by aspirin intake, since this study recorded aspirin use (defined as twice per week for at least a year) [42,43,44]. This study comprised 4410 cases, of which 810 (18.37% of cases) were aspirin users and 3340 (75.74%) were non-users, and 260 cases (5.90%) were excluded as they had reported use of other non-aspirin NSAIDs. This study also contained 3441 controls, of which 779 (22.64%) had recorded aspirin use and 2320 (67.42%) were recorded as non-users, and 342 controls (9.94%) were excluded as they had reported use of other non-aspirin NSAIDs. 

To assess the causal effect of SA on CRC risk, we tested for association in GECCO, and also stratified the analysis between aspirin users and non-users in DACHS to investigate whether increased SA levels via pharmacological intervention is required to see an effect. We obtained summary association statistics from GECCO and also conducted logistic regression analyses adjusting for age and sex in the DACHS study for all the participants. We then stratified the participants of the DACHS study to aspirin users and non-users before repeating the logistic regression analyses again. Genetic instruments that had a MAF ≤ 0.01 in both GECCO and DACHS (all participants) were excluded from further analyses.

### 2.3. Statistical Analyses

Analyses were carried out in R version 3.2.3 using the “Two-Sample MR” package [45]. This package allows the formatting, harmonisation, and analysis of summary data from genetic association studies in a semi-automated manner. The Two-Sample MR package automatically assigns effect alleles so that SNP associations with the exposure are positive i.e., so the effect allele is “metabolite-increasing”. The SNPs identified as associated with SA can then be extracted from the outcome datasets. Allele harmonization ensures that the effect (metabolite-increasing) allele in the exposure dataset is also treated as the effect allele in the outcome dataset. When only one SNP was associated with the metabolite, Wald ratios (SNP–outcome estimate ÷ SNP–exposure estimate) were calculated to assess the change in log OR per SD increase in the metabolite. When more than one SNP was available, a weighted mean weighted by the inverse variance of the Wald ratio estimates (inverse-variance weighted (IVW) method) was used to assess the causal effect of increased metabolite levels on risk of CRC incidence [46]. To assess the quality of our instruments, we calculated the variance in SA levels explained by the SNPs and the F statistic. The variance explained for each SNP was calculated using the formula: 
$\frac{2b^{2}p\;\left(1-p\right)}{var}$
, where *p* is the minor allele frequency, *b* is the SNP effect on the exposure (beta) and *var* is the variance of the exposure. The F statistic was calculated using the formula: 
$\frac{r^{2}\left(n-1-k\right)}{\left(\left(1-r^{2}\right)k\right)}$
 where *r* is the sum of the variance explained by the set of SNPs, *n* is the sample size of the exposure GWAS and *k* is the number of SNPs used to proxy the exposure. In the presence of weak instruments, we conducted an MR robust adjusted profile score (MR RAPS), which is a method that provides robust inference when many weak instruments are present [47].

Furthermore, the presence of one invalid instrument, e.g., one that is associated with exposures other than the exposure of interest (horizontal pleiotropy), may bias the results from the IVW method [48]. For this reason, alternative methods that produce an unbiased estimator even when some of the genetic instruments are invalid were used as a sensitivity analysis when more than 2 SNPs were used as exposure instruments (weighted mode, weighted median, and MR Egger) [45,49,50,51]. The MR Egger test is not constrained to pass through an effect size of 0, unlike the IVW method, allowing the assessment of the presence of directional pleiotropy through the y intercept [48,51]. We also measured the Q statistic to measure the presence of pleiotropy between our instruments. If all the SNPs are valid instruments, then the individual MR estimates for each SNP will only vary by chance. A larger amount of heterogeneity would indicate that one or more of the SNPs are pleiotropic [52].

Due to the presence of a small number of independent SNPs associated with the metabolite, we also conducted a weighted generalised linear regression (WGLR), whereby SNPs in LD (R^2^ < 0.8) could be used with the incorporation of their correlation as weights in the regression analysis [53]. This was performed using the “LDlinkR” and “MendelianRandomization” packages in R (version 3.5.1). The use of multiple SNPs explains more of the variance in the metabolite levels and therefore improves power to detect an effect [53]. 

We also assessed the possibility of reverse causation through the use of the MR Steiger test, found in the “TwoSampleMR” package, which does so by comparing the variance explained by the SNPs for the exposure and the outcome [45].

## 3. Results

### 3.1. Functional SNPs and CRC Risk

To interrogate the effect of SA on CRC risk, we used three methods to select our exposure instruments (Figure 2). In our first approach, we identified four functional SNPs that have been shown to affect enzyme efficiency in the aspirin metabolic pathway (Figure 1). For BChE (rs6445035), the presence of an A allele increase has been associated with a decrease in aspirin hydrolysis by around 1.2 nmol/mL/min [54]. The UGT1A6 variants, rs2070959 and rs1105879, predict a higher metabolic activity of the enzyme than the wild type [55,56]. Furthermore, a variant in CYP2C9 (rs1799853) encodes an enzyme with reduced activity [57]. 

These SNPs were tested for association with SA in the INTERVAL and (EPIC)-Norfolk study, however none of the SNPs reached nominal significance with the metabolite (Figure 3A) (Appendix A). For this reason, these SNPs were therefore not taken forward in an MR analysis. 

### 3.2. Pathway SNPs and CRC Risk

We investigated genetic variants within the coding regions of the enzymes involved in aspirin and SA metabolism (Figure 1). These were BChE, PAFAH1b2, PAFAH1b3, UGT1A6, ACSM2B and CYP2C9. 

We obtained summary statistics for 2701 SNPs within the genetic coding regions of the enzymes for SA. We applied a Bonferroni threshold of association (P value 0.05/2701 = 1.85 × 10^−5^) for SNPS and restricted to SNPs with a consistent direction of effects in both studies and a minor allele frequency of ≥0.01 in the exposure and outcome studies. This identified 45 SNPs that could be used to instrument SA. These SNPs were then clumped at an R^2^ < 0.001 and 0.8, providing two and six SNPs, respectively, to instrument SA levels (Figure 2). These explained 0.03% and 0.09% of the variance in SA levels, and had an F statistic of 1.74 and 2.16, respectively (Table 1). 

After LD clumping at an R^2^ < 0.001, 2 SNPs were taken forward in an IVW analysis. We found little evidence of an association between an SD increase in SA and CRC risk (GECCO OR: 1.03, 95% CI: 0.84–1.27 and DACHS OR: 1.10, 95% CI: 0.58–2.07) (Figure 3B). Since aspirin is rapidly deacetylated to form SA [22] and therefore a plausible proxy for increased SA levels, we stratified our analysis between aspirin users and non-users in the DACHS study. Our power calculations show that after stratification, we had 80% power to detect an effect of an SD increase in SA on CRC risk with an OR of ≤0.43 and ≥2.38 in the reciprocal direction for aspirin users (*n* = 1589). For non-users (*n* = 5660), we had 80% power to detect an OR of ≤0.64 and ≥1.64 in the reciprocal direction (Table 1). However, our MR analysis showed little evidence of an association between SA and CRC risk (OR: 0.93, 95% CI: 0.23–3.73 and OR: 1.24, 95% CI: 0.57–2.69, respectively) (Figure 3B). 

The variance explained by these two instruments and their F statistics indicate the possibility of weak instrument bias. For this reason, we conducted MR RAPS, a method that provides robust inference even in the presence of weak instruments [47]. Through this method, no association was found between an SD increase in SA and CRC risk (GECCO OR: 1.04, 95% CI: 0.87–1.23 and DACHS OR:1.10, 95% CI: 0.57–2.12). Results remain unchanged, even after stratification between aspirin users and non-users (OR: 0.93, 95% CI: 0.22–3.87 and OR: 1.24, 95% CI: 0.56–2.76).

Since this LD threshold is known to be very stringent, we used a more relaxed threshold (R^2^ < 0.8) to increase the number of SNPs available to instrument the metabolite and therefore explain more of the variance in SA levels. This provided six SNPs associated with SA shown in Appendix A, of which SNP associations with the outcome are also provided in Appendix A. These SNPs showed no association between SA and CRC risk (GECCO OR: 1.01, 95% CI: 0.91–1.12 and DACHS OR:1.14, 95% CI: 0.77–1.68). Stratification between aspirin use and non-use found no association between the metabolite and CRC risk in aspirin users or non-users (OR: 1.02, 95% CI: 0.44–2.40 and OR: 1.26, 95% CI: 0.78–2.01, respectively). 

Using the alternative MR methods (weighted mode, weighted median, and MR Egger), no other association between SA and CRC in both GECCO and DACHS was observed, regardless of stratification (Appendix A). 

Since all the SNPs were found to be on chromosome 16 (Appendix A), a WGLR method was carried out to account for the SNP correlations and include them as weights into the regression. Through this method, there was no association between SA and CRC risk in DACHS (OR: 0.81, 95% CI: 0.36–1.83) but a positive association in the GECCO sample (OR: 1.11, 95% CI: 1.01–1.21). No association was observed between SA and CRC risk in aspirin users or non-users (OR: 0.35, 95% CI: 0.05–2.47 and OR: 1.10, 95% CI: 0.55–2.16, respectively) (Figure 3B). As a sensitivity analysis, the heterogeneity of the results was appraised through a Q statistic, but no evidence of pleiotropy was observed- i.e., no evidence that the instruments may also be associated with another phenotype (Appendix A). 

To identify whether the causal pathway was in the direction from SA to CRC and not the reverse, we performed the MR Steiger method using the functional SNPs [58]. This method suggested that the causal direction was indeed from SA to CRC, because the SNPs explained more variation in SA levels than CRC risk (Appendix A).

### 3.3. Genome-Wide Significant SNPs and CRC Risk

Initially, 72 SNPs were associated with SA at genome-wide significance. We restricted our analysis to SNPs with a MAF threshold of ≥0.01 in the exposure and outcome studies, and only included those with a consistent direction of effect in both studies. This resulted in 58 SNPs that were available to instrument SA. After removing SNPs in LD at an R^2^ < 0.001 and R^2^ < 0.8, one SNP and four SNPs were available to instrument SA, respectively (Figure 2). These explained 0.05% and 0.09% of the variance in SA levels and had an F statistic of 7.44 and 3.18, respectively (Table 1). 

Using the one independent SNP associated with SA at genome-wide significance, WR results showed no association between the genetically predicted metabolite levels and cancer risk (GECCO OR: 1.08, 95% CI: 0.86–1.34 and DACHS OR: 1.01, 95% CI: 0.44–2.31). Our power calculations show that after stratification between aspirin users and non-users in the DACHS study, we had 80% power to detect an effect of an SD increase in SA on CRC risk with an OR of ≤0.55 and ≥1.83 in the reciprocal direction for aspirin users (*n* = 1589). For non-users (*n* = 5660), we had 80% power to detect an OR of ≤0.73 and ≥1.42 in the reciprocal direction (Table 1), however, we found no association between SA levels and CRC in aspirin users (OR: 0.66, 95% CI: 0.11–4.12) and non-users (OR: 1.12, 95% CI: 0.42–2.97) (Figure 3C). 

Due to the possibility of weak instrument bias, we also conducted an MR RAPS approach, but results remained unchanged (GECCO OR: 1.08, 95% CI: 0.86–1.36, DACHS OR: 1.01, 95% CI: 0.44–2.36, DACHS aspirin users OR: 0.66, 95% CI: 0.10–4.33 and DACHS aspirin non-users OR: 1.12, 95% CI: 0.41–3.04). 

To explain more of the variance, we used a less stringent LD threshold of R^2^ < 0.8, and therefore four SNPs to instrument SA (Appendix A, associations with CRC are also found in Appendix A). IVW results also showed no association between the metabolites and CRC risk (GECCO OR: 1.03, 95% CI: 0.92–1.15, and DACHS OR: 1.06, 95% CI: 0.69–1.63) and no association was found upon stratification by aspirin use (users OR: 0.99, 95% CI: 0.38–2.57, non-users OR: 1.10, 95% CI: 0.66–1.84). 

Using the alternative MR methods (weighted mode, weighted median, and MR Egger), no association between SA and CRC in both GECCO and DACHS was seen, regardless of stratification (Appendix A). 

Since these four SNPs were all found on chromosome 16 (Appendix A), a WGLR method was applied to account for their correlation. We found a positive association between SA and CRC risk in the GECCO sample (OR: 1.13, 95% CI: 1.05–1.22) but no association in the DACHS sample (OR: 0.51, 95% CI: 0.16–1.67), DACHS aspirin users (OR: 0.12, 95% CI: 0.01–2.67) and DACHS aspirin non-users (OR: 0.70, 95% CI: 0.30–1.65) (Figure 3C). As a sensitivity analysis, the heterogeneity of the results was assessed through a Q statistic, but no evidence of heterogeneity was seen (Appendix A). 

We repeated the MR Steiger method using the genome-wide significant SNPs, and results again suggested that the causal direction was indeed from SA to CRC, as the SNPs explained more variation in SA levels than CRC risk (Appendix A).

## 4. Discussion

In this study, we aimed to assess whether increasing levels of SA affected risk of CRC, using an MR approach, and whether higher levels of SA proxied by pharmacological intervention in the form of aspirin use was required to identify an effect. Our analysis focused on aspirin, since 90% of the drug is rapidly deacetylated to form SA [16], which is the active metabolite of the drug [13,14], and therefore increases SA levels more than would be achieved through the diet. Three different approaches were applied to identify genetic variants (instrument variables) which could serve as proxies for SA and understand the causal nature of their role in determining CRC risk. The three approaches involved selecting (i) functional, (ii) pathway and (iii) genome-wide SNPs each associated with SA. The functional genetic variants were selected through the established role of the genes in aspirin metabolism from various sources of evidence. With regards to the pathway and genome-wide significant SNPs, all were found on chromosome 16, either within or near the coding region for the enzyme ACSM2B, which is the enzyme involved in breaking down SA into its metabolite salicyluric acid, thereby providing a plausible biological link between these SNPs and levels of SA. 

We found no association between the functional SNPs and levels of SA, therefore did not take them forward to instrument SA levels. Using pathway and genome-wide SNPs, we identified two and one independent SNPs (R^2^ < 0.001) to proxy for SA levels, respectively, and found no association between increasing metabolite levels and CRC risk using an IVW and MR RAPS approach, regardless of aspirin stratification. Furthermore, due to the small number of instruments, we applied a less stringent LD threshold (R^2^ < 0.8) and identified six pathway SNPs and four genome-wide SNPs to proxy for an SD increase in SA levels. Using these SNPs, we found consistent null results using the IVW method and alternative MR methods (weighted median, weighted mode, and MR Egger). However, after accounting for SNP correlation using a WGLR method, we found that an SD increase in SA increased the risk of CRC in GECCO (OR: 1.11, 95% CI: 1.01–1.21, *p*-value: 0.03 and OR: 1.13, 95% CI: 1.05–1.22, *p*-value: 1.42 × 10^−3^, respectively). We acknowledge that when the LD clumping threshold was relaxed to 0.8, there may have been some overlap with SNPs used in both the functional and the genome-wide analysis. We also acknowledge that the sensitivity analyses used were limited in detecting heterogeneity due to the low number of SNPs. The Cochran Q statistic also requires a large number of SNPs, otherwise there is little power to detect heterogeneity [59]. Overall, we found little evidence to suggest that SA affects risk of CRC, regardless of stratification. 

It is thought that one reason why fruit and vegetable consumption may prevent CRC [8] is due to the presence of SA [60], although no formal RCTs have been carried out to confirm this. In vitro studies have also shown that salicylic acid inhibits the growth of colorectal cancer cells [61]. SA is the primary metabolite of aspirin, of which both observational and RCT evidence have shown aspirin as a chemopreventive agent [18,19,20,21]. Our MR results show little evidence of an association between the metabolite and CRC risk, regardless of aspirin use. However, we discuss some of the possible reasons why below. 

Whilst we found no association between functional SNPs known to affect aspirin metabolism enzymes’ activity and levels of SA, this may be due to a more complex relationship between genotype and metabolite levels, rather than the assumed linear additive model. For example, with regards to the functional SNPs, Nagar et al. (2004) identified that whilst individuals with homozygous mutant alleles of UGT1A6 had the highest metabolic activity, those that were heterozygous for alleles in three SNPs (including rs1105879 and rs2070959) were actually less active than homozygous wildtype enzymes [56], indicating a non-linear association between the alleles and the metabolites, which is a common assumption made in regression analyses [62]. This non-linear association between alleles and enzyme activity needs to also be addressed between alleles and metabolite levels to derive instrumental variables. 

To our knowledge, our GWAS for SA is the largest performed for this metabolite (*n* = 14,149), with others having much smaller sample sizes and not being publicly available. By using a much larger sample size, we were able to identify genome-wide significant associations that would have otherwise been missed in smaller studies. However, the variance explained and the strength of the instruments still indicated weak instruments despite strong associations with the metabolite. In order to improve the results and conclusions observed in this study, ideally we would need to identify the SNP associations with SA levels stratified between aspirin users and non-users, similar to what was carried out in our CRC outcome sample. However, to our knowledge, metabolite, genotype, and phenotype data (of aspirin use) are not currently large enough to run this analysis. If a stronger association exists between the SNPs and SA levels in aspirin users, this would provide more strength of the appropriateness of the genetic instruments used to proxy for SA levels. 

We also acknowledge another limitation in this study is that the measurement of metabolites was through an untargeted metabolomics approach, and so the variables generated are assessed in units of measurement called “ion counts” which are calculated from the area under the curve of the corresponding peak in the mass spectrum. This means that metabolite measurements are quantitative values of relative changes as opposed to the absolute quantification of metabolite concentrations that can be achieved through targeted metabolomics [63]. For this reason, it is important to focus on the direction of effect and strength of association (*p*-values) in this study, as opposed to the magnitude of effect. This may have also impacted on the calculation of variance explained and the F statistics, which mostly indicate that the instrumental variables used in the MR were weak as they explain little of the variance, and the F-statistic is below the conventionally applied indicative threshold of 10 [64]. However, without carrying out a more targeted metabolomic approach and quantifying the exact effect of these SNPs on the metabolite levels, it is difficult to draw firm conclusions about the strength of the instruments used for MR. 

Furthermore, larger sample sizes of recorded aspirin use are required as currently, our study may have been underpowered to detect an effect, hence explaining the null results using the IVW approach. Our power calculations show that in aspirin users, we had sufficient power to detect an effect of SA on CRC risk with an OR of ≤0.43 and ≥2.38, whereas observationally, the effect of aspirin on CRC risk is RR: 0.79 (95% CI: 0.74–0.85) [19], and in a long-term observational follow-up of a trail, the hazard ratio was 0.80 (95% CI: 0.67–0.97) [18]. Therefore, it would be useful to repeat this analysis in a larger sample with comprehensive data on aspirin use.

## 5. Conclusions

Overall, the analyses presented have shown that dietary levels of SA, as well as increased levels proxied by aspirin use, may be insufficient at reducing risk of CRC, although based on the variance explained in SA levels by our SNPs and the F statistic, we acknowledge that the analysis needs to be repeated again with stronger instruments that proxy the metabolite levels.

## Figures and Tables

**Figure 1 nutrients-13-04164-f001:**
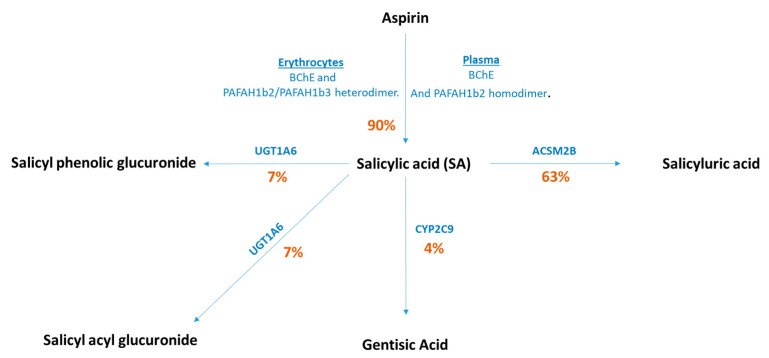
Aspirin metabolism pathway. Roughly 10% of aspirin remains unchanged and is excreted in the urine as aspirin. Aspirin is broken down into various metabolites, the most active of them being salicylic acid [14,16]. Various enzymes are involved in the metabolism pathway. The percentages indicate how much of the drug is being metabolised in that pathway. Abbreviations: BChE, butyrylcholinesterase; PAFAH1b2, platelet-activating factor acetylhydrolase 2; PAFAH1b3, platelet-activating factor acetylhydrolase 3; UGT1A6, UDP-glucuronosyltransferase 1–6; ACSM2B, Acyl-CoA Synthetase Medium-Chain Family Member 2B and CYP450, cytochrome P450.

**Figure 2 nutrients-13-04164-f002:**
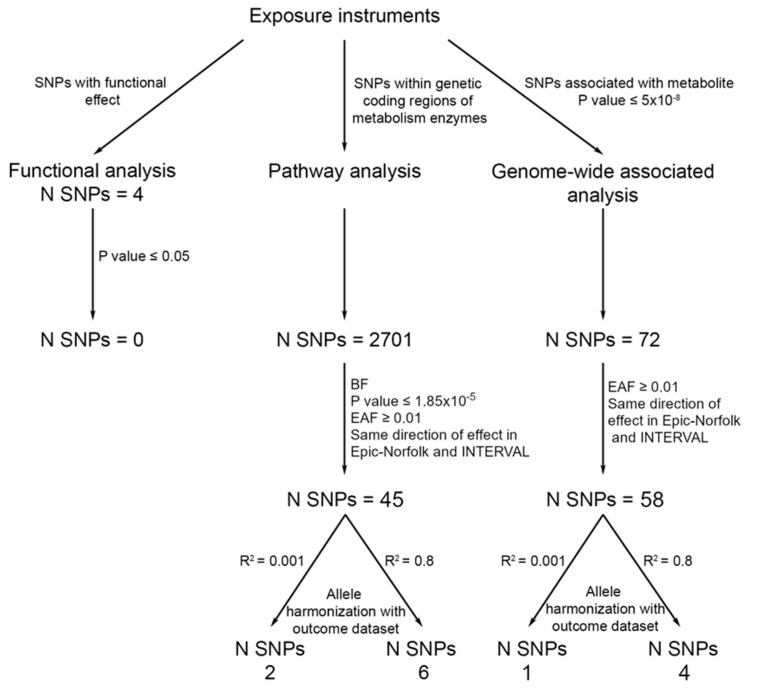
Instrument selection for functional, pathway, and genome-wide SNPs. Abbreviations: SA, salicylic acid; EAF, effect allele frequency; BF, Bonferroni.

**Figure 3 nutrients-13-04164-f003:**
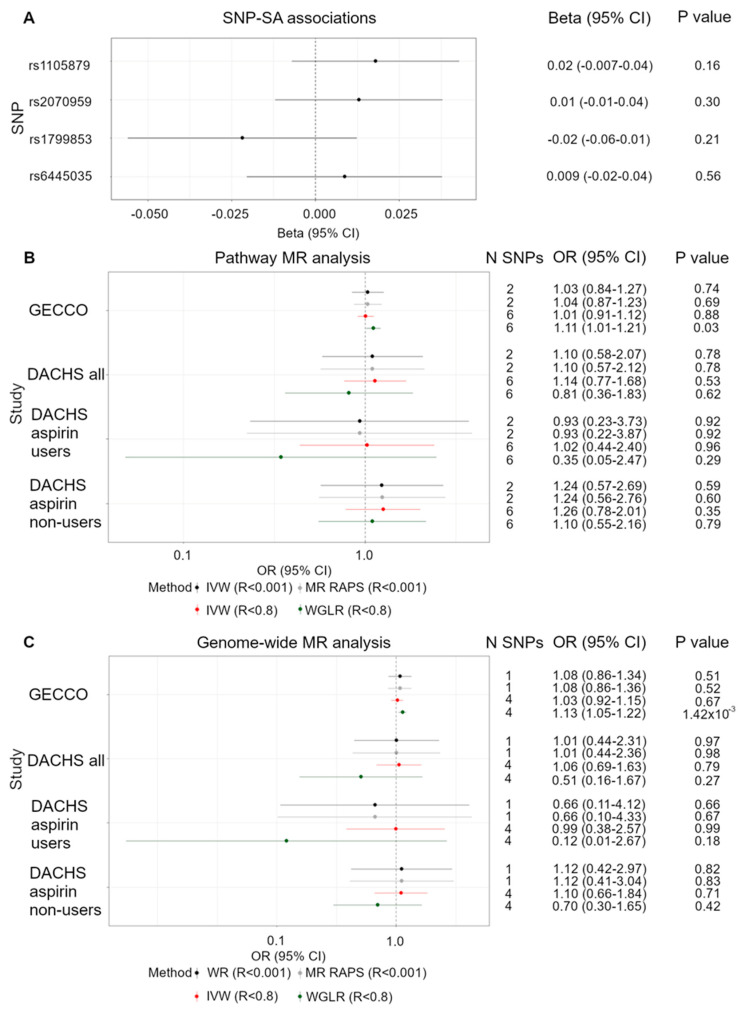
Functional SNP metabolite associations and two-sample pathway MR analysis. (**A**) Forest plot of single SNP associations with salicylic acid for the functional SNPs. (**B**) Forest plot of one SD increase in SA and its effect on CRC risk, instrumented by pathway SNPs and applying three methods: IVW after applying an LD threshold of R^2^ < 0.001 (black), MR RAPS after applying an LD threshold of R^2^ < 0.001 (grey), IVW after applying an LD threshold of R^2^ < 0.8 (red) and a WGLR after applying an LD threshold of R^2^ < 0.8 (green). (**C**) Forest plot of one SD increase in SA and its effect on CRC risk, instrumented by genome-wide SNPs and applying three methods: WR after applying an LD threshold of R^2^ < 0.001 (black), MR RAPS after applying an LD threshold of R^2^ < 0.001 (grey), IVW after applying an LD threshold of R^2^ < 0.8 (red) and a WGLR after applying an LD threshold of R^2^ < 0.8 (green). Abbreviations: OR, odds ratio; IVW, inverse variance weighted; WGLR, weighted generalised linear regression; WR, Wald ratio; LD, linkage disequilibrium.

**Table 1 nutrients-13-04164-t001:** Exposure instruments used in the MR analysis.

SNP Set	Study	Outcome Sample Size	Percentage Cases (%)	N SNPs	LD R^2^	Variance Explained R^2^ (%)	F Statistics	OR Detected at 80% Power	
								Decreased Risk	Increased Risk
Pathway SNPs	GECCO	120,328	45.85 55,168/120,328)	2	0.001	0.025	1.74	0.90	1.11
	DACHS	7851	56.17	2				0.68	1.51
	DACHS aspirin users	1589	(4410/7851)	2				0.43	2.38
	DACHS aspirin non-users	5660	50.98	2				0.64	1.64
	GECCO	120,328	(810/1589)	6	0.8	0.092	2.16	0.95	1.06
	DACHS	7851	59.01	6				0.81	1.24
	DACHS aspirin users	1589	(3340/5660)	6				0.63	1.58
	DACHS aspirin non-users	5660	45.85 (55,168/120,328)	6				0.78	1.30
Genome-wide SNPs	GECCO	120,328	45.85 55,168/120,328)	2	0.001	0.053	7.44	0.93	1.07
	DACHS	7851	56.17	2				0.76	1.32
	DACHS aspirin users	1589	(4410/7851)	2				0.55	1.83
	DACHS aspirin non-users	5660	50.98	2				0.73	1.42
	GECCO	120,328	(810/1589)	6	0.8	0.090	3.18	0.95	1.06
	DACHS	7851	59.01	6				0.81	1.24
	DACHS aspirin users	1589	(3340/5660)	6				0.63	1.59
	DACHS aspirin non-users	5660	45.85 (55,168/120,328)	6				0.78	1.30

Abbreviations: SA, salicylic acid; LD, linkage disequilibrium; NA, not applicable; OR, odds ratio.

## Data Availability

Summary data is available upon request by contacting the respective studies.

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
