# Peer review of "Salicylic Acid and Risk of Colorectal Cancer: A Two-Sample Mendelian Randomization Study"

_nutrients, 2021, doi:10.3390/nu13114164_

Round 1
Reviewer 1 Report
This is a well-written manuscript about a 2-sample Mendelian Randomization (MR) study to investigate the causal effect of salicylic acid (SA) on CRC risk. Authors describe in the introduction that SA is a metabolite that can be derived from food (fruits, vegetables) or from pharmaceutics such as aspirin. There is good evidence that aspirin use is associated with lower risk of CRC, while fruits and vegetables intake has also been associated with lower CRC risk but whether this is due to SA is questionable. Authors apply advanced MR methods thoughtful to answer their research question. Unfortunately, only very small proportions of interindividual SA variation are explained by the identified genetic variants and overall no association between genetically determined higher SA and CRC risk was observed in the overall GECCO dataset and also not in the smaller DACHS dataset, where they could also stratify by aspirin use. My few comments are as follows:
(1) It may be helpful to the readers to clearly state the hypothesis and the direction of the hypothesized association, i.e. given the inverse association between Aspirin use and CRC was hypothesized that genetically determined higher SA is associated with lower CRC risk? Also, authors may describe more clearly what this MR adds given the good evidence on aspirin use and CRC risk. If they are thinking in the direction of personalized prevention, wouldn’t it be more straightforward to stratify a large study or cohort consortium on aspirin use and CRC risk by genetic variation of SA metabolism?
(2) In the introduction, authors state that fruits and vegetables intakes are inversely associated with CRC risk, while the exact components underlying this effect are unknown. However, there is quite good evidence for certain components such as dietary fiber and lower CRC risk, which should be acknowledged here.
(3) Authors conducted their own GWAS on SA levels using data from EPIC-Norfolk and INTERVAL. It should be briefly commented on this in the discussion Were there no other GWAS available? Are there candidate gene studies on circulating SA? Are there advantages / disadvantages of this approach compared to using published GWAS data?
Author Response
Thank you for your comments- I will address these below:
1- hypothesis of inverse association has been added on line 224-226 in the tracked document and untracked document.
we have also added a couple of lines at 201-203 of the tracked document and line 200-202 in the untracked document - the purpose of the study was to investigate SA itself using MR, not aspirin use. This is because the evidence between aspirin and CRC itself is clear. We wanted to see whether using MR would help identify an association between SA and CRC in a much quicker manner
2- Yes agreed, I have edited the lines 166-170 of the tracked document and lines 166-169 of the untracked document to reflect this.
3- I have added in a few sentences in the discussion section about this - lines 488-493 in the tracked document and lines 487-492 of the untracked document.
Reviewer 2 Report
In this article, the authors used Mendelian randomization methods to evaluate the effect of salicylic acid in colorectal cancer. The article is well written in general, though there are a few minor mistakes such as the repetition of the word – “the” - in line 489 and a few grammatical errors/typos. More importantly, below are some of the major comments that I hope the authors find useful in their potential revision of this work.
Minor:
- In line 311 the concept of weak instrumental variables is used, however, is not defined until line 499.
- Figure 2 could be a better introduction to the approaches used in the method section, instead of being the first part of the results.
- The way the paragraphs are structured in the results section make them hard to read, as several paragraphs seemed to only contain one long sentence.
- Header/title for Table 1 is apparently missing from the text.
There are also some points that will need to be addressed: -
- The authors present three different processes for the selection of instrumental variables; functional, pathway and genome-wide SNP’s. However, it might be helpful for the authors to clarify in text that these approaches are not completely independent as instrumental variables/SNPs selected through different approaches might overlap or be in close LD.
- It is not clear which method was used for clumping, and the WGLR method is not defined or explained in the manuscript. More details need to be provided either in the methods section or in the supplementary material document.
- Is not completely clear how the author's tests for pleiotropy: e.g. were other traits used as negative/positive controls? If so, these need to provided as part of supplementary materials. Moreover, MR tests for directionality such as the MR-Steiger test can be used here to test for direct association between the genetic instrumental variables and colorectal cancer (indep of SA).
- Discussion on the limitation of some of the adopted sensitivity MR analysis seems lacking. It is important to acknowledge that the ability to assess for pleiotropy is limited due to the low number of SNP’s adopted in this study. A quick get-around on this, would be to potentially relax the significant threshold for SNP inclusion (i.e. for instrument selection) on SA, and re-evaluate findings from various MR sensitivity analysis models to harvest better power.
- An alternative approach due to the limited number of SNP’s could be to use other methods such as polygenic risk scores PRS to test if highly associated SNP’s with salicylic acid can predict colorectal cancer. Given the availability of independent datasets.
- The discussion at its current form make very little mention on putting previous findings in the context of this study, or to provide reader some hint on how this study adds onto the existing body of evidence. There is also some observed conflicting direction of effect between MR-derived SA-CC relationship (GECCO) and those derived through earlier observational studies (in the meta-analysis of cohort studies) that were not explained in the main text. Perhaps some clarification should be made on whether MR were able to support/refute the existing effect sizes observed in earlier observational findings for SA (or related traits) and colorectal cancers. Based on the power calculation, it gives a rather weak impression that MR did not have sufficient power to support nor reject the observational RR of 0.9 (which is approximately ~OR 0.9 since CC is relatively rare), so more justification/explanation on this matter might make the overall message of the paper clearer.
- Finally, to ensure replicability of these findings by other researchers, I urge the authors to provide the individual SNP estimates (SNP RSID, EAF, EA, OA, Beta, se, p) for each SA SNPs used in the present study on CC risk. Given that the list of SNPs used were not exhaustive, it might suffice to provide them through supplementary tables within Supp materials.
Author Response
Please find it in the attachment. Thank you.

Reviewer 3 Report
The authors aimed to address whether levels of SA affected CRC risk, stratifying by aspirin use. The methods are appropriate and the introduction, results, and discussion are well written.
Author Response
Thank you for your comments.
As per what I have understood, there is nothing for me to address/change in the paper
Round 2
Reviewer 2 Report
I appreciate the authors' effort to revise their manuscript in a timely manner and made necessary corrections. The paper is now much easier to follow.
I have two remaining points to raise:
1) the discussion need to acknowledge the limitation of sensitivity analyses present in this MR study to detect/correct for pleiotropy given the very low number of SNPs. The same caveat applies for Cochran Q stats. Currently limitations on sample sizes have been provided, but it is crucial to emphasize the limitation of existing stat-based sensitivity analyses models when dealing with less than 5 SNPs.
2) My apologies for the unclear instruction previously. I previously meant that the SNP-level association statistics between SA SNPs and CRC risk need to be provided/made available for replicability (i.e. the SNP-CRC betas, not just the SNP-SA betas).
Author Response
1- I've included a couple of sentences in lines 506-509 reflecting that we had few SNPs to truly test heterogeneity
2- I have added in SNP associations with CRC in Supplementary table 4 and 9